# Integrating Multi-Omics Analysis Reveals the Regulatory Mechanisms of White–Violet Mutant Flowers in Grape Hyacinth (*Muscari latifolium*)

**DOI:** 10.3390/ijms24055044

**Published:** 2023-03-06

**Authors:** Junren Ma, Zhi Li, Yali Liu

**Affiliations:** 1College of Landscape Architecture and Arts, Northwest A & F University, Xianyang 712100, China; 2State Key Laboratory of Crop Stress Biology for Arid Areas, College of Horticulture, Northwest A & F University, Xianyang 712100, China; 3Key Laboratory of Horticultural Plant Biology and Germplasm Innovation in Northwest China, Ministry of Agriculture, Xianyang 712100, China

**Keywords:** grape hyacinth, mutant, flower color, ionomics, metabolomic, transcriptomic, MYB

## Abstract

Grape hyacinth (*Muscari* spp.) is a famous bulbous blue flower; however, few bicolor varieties are available in the market. Therefore, the discovery of bicolor varieties and understanding of their mechanisms are crucial to the breeding of new varieties. In this study, we report a significant bicolor mutant with white upper and violet lower portions, with both parts belonging to a single raceme. Ionomics showed that pH and metal element contents were not responsible for the bicolor formation. Targeted metabolomics illustrated that the content of the 24 color-related compounds was significantly lower in the upper part than that in the lower part. Moreover, full-length transcriptomics combined with second-generation transcriptomics revealed 12,237 differentially expressed genes in which anthocyanin synthesis gene expression of the upper part was noted to be significantly lower than that of the lower part. Transcription factor differential expression analysis was used to describe the presence of a pair of *MaMYB113a/b* sequences, with low levels of expression in the upper part and high expression in the lower part. Furthermore, tobacco transformation confirmed that overexpression of *MaMYB113a/b* can promote anthocyanin accumulation in tobacco leaves. Accordingly, the differential expression of *MaMYB113a/b* contributes the formation of a bicolor mutant in *Muscari latifolium.*

## 1. Introduction

Color is considered to be one of the most important ornamental traits of flowers. Each ornamental color confers a considerable influence on the human soul and emotional perception in the industries of horticulture, forestry, and agriculture. In nature, many flowers are of a single color. Therefore, in order to satisfy consumer preference, the discovery and cultivation of varieties with two or more colors has become the focus of researchers and breeders [1].

The vast majority of flower color formation is the result of the accumulation of certain compounds, which can be divided into four main categories: chlorophylls, carotenoids, betalains, and flavonoids. Flavonoids are the most widespread and affect flower color, with more than 9000 types existing in the plant kingdom [2,3]. This pathway is sequentially catalyzed and modified by chalcone synthase (CHS), chalcone isomerase (CHI), naringenin 3–dioxygenase (F3H), flavonoid 3’–hyroxylase (F3’H), flavonoid 3’, 5’–hydroxylase (F3’5’H), dihydroflavonol 4 reductase (DFR), anthocyanidin synthase (ANS), and glycosyltransferase (GT) to form anthocyanin and derivatives, such as pelargonidin, pelargonidin–3–*O*–glucoside, cyanidin, cyanidin–3–*O*–glucoside, delphinidin and delphinidin–3–*O*–glucoside [4,5].

Studies have shown that the synthesis of flower pigments is regulated by transcription factors [6]. V-myb avian myeloblastosis viral oncogene homolog (MYB), basic helix–loop–helix (bHLH), and wd40–repeat protein (WD40) are known to be the most widely studied transcription factors [7]. The regulation of MYB transcription factors, especially R2R3–MYB, is also considered to be essential, and can solely form MYB–bHLH (MB), MYB–WD40 (MW) or MYB–bHLH–WD40 (MBW) protein complexes to modulate the expression of structural genes through various interactions [8].

Furthermore, the final formation of flower color has been shown to be influenced by many physiological and biochemical factors, such as vacuole pH [9,10] and metal element content [10,11,12].

Grape hyacinth (*Muscari* spp.) is a famous bulbous, blue-flowered plant that is widely used as ground cover, as well as potted and cut flowers. There are approximately 77 species of *Muscari* worldwide, of which 51 species are found in Turkey, with 34 unique to the region [13]. *Muscari latifolium* has a unique blue–violet inflorescence and broad leaves. It is endemic to Turkey and is distributed in the pine forests of the western, central–western, and southwestern Balıkesir–Çanakkale Kazda Mountains, Kütahya Murat Mountain and Antalya Akseki at an altitude of 1100–1800 m [14]. In previous studies, Mori applied high-pressure liquid chromatography (HPLC) to separate and compare blue and violet tepals; the upper blue flower was speculated to be mainly delphinidin, whereas the lower blue–purple flower was a mixture of cyanidin and delphinidin [15]. Based on HPLC and qRT–PCR, Qi believed that—owing to the differential expression of *F3’5’H, F3’H* and *DFR*—the upper blue flowers only contained delphinidin, whereas the lower purple flowers had a more mixed accumulation of cyanidin and delphinidin, although these elements and pH were not found to be the cause of the color change [16]. Wang and Lou reached different conclusions via HPLC and untargeted mass spectrometry. Specifically, they reported that the upper blue flowers did not contain delphinidin its derivatives but contained pelargonidin and cyanidin derivatives [17,18]. Because the exact mechanism of color formation in *M. latifolium* has yet to be studied in detail by full-length transcriptomics and targeted metabolomics, the function of its flower color transcription factors remains unknown. Another study revealed that in the cultivar *M. armeniacum, MaMYBAN2*, and *MaMybA* can promote the accumulation of anthocyanin in tobacco [19,20].

In this study, a comparative analysis of the integrating ionomics, targeted metabolomics, and full-length transcriptomics of new white–violet mutant flowers were conducted, in which a pair of *MaMYB113a/b* transcription factors was identified, which was significantly downregulated in the upper white flowers and upregulated in the lower violet flowers. Moreover, analysis of transgenic tobacco shows that a high level of expression of *MaMYB113a/b* can promote tobacco anthocyanin accumulation. Accordingly, this study can serve as a reference for breeding and in attempts to elucidate the formation mechanism of double-color flowers.

## 2. Results

### 2.1. Phenotypic and Ionomics Analysis of M. latifolium White Flower and Violet Flower

The results of the phenotypic analysis of the white–violet bicolor inflorescence mutant material are shown in Figure 1. The white and violet tepals were found to differ significantly, with no pigment accumulation observed in the white tepals. In the violet tepals, the pigment accumulated in the uppermost palisade tissue cells, adjacent to the epidermal cells. Moreover, an in vivo puncture pH test of the white and violet tepals demonstrated that both white and violet tepals had pH value of less than 6, with no difference in the significance test. Twelve metal elements that play a key role in the formation of blue–purple flowers, which were summarized in a review article, were selected for ICP–MS analysis [9]. Both white and violet flowers were found to contain a large amount of Mg, Al, Ca, Mn, Fe, and Zn (Figure 1I; Appendix A). Although the average of each element differed between white and violet flowers in terms of absolute value, no difference was observed in the overall multivariate significance Hotelling’s *T^2^*-test analysis of the twelve elements. According to the review summary, the presence of several large elements in the 12 metal elements was clearly shown to be sufficient to form flower pigment complexes [9]. Therefore, the formation of *M. latifolium* white flower pigments was not the result of a lack of certain metal elements or the change in pH.

### 2.2. Targeted Metabolomics Analysis of White and Violet Flowers of M. latifolium

Using 72 flower color related standards, white and violet flowers were analyzed by UPLC–QQQ–MS. The detailed detection parameters for each standard are shown in Appendix A. Here, a total of 37 compounds were detected in both white and violet flowers. The white flower was found to contain 28 compounds, while the violet flower contained 34 compounds, with 25 compounds being common to both. The statistics of differentially expressed compounds demonstrated that 24 compounds in violet flowers were noted to be significantly increased. Among these 24 compounds, half were anthocyanins. Specifically, the anthocyanin response intensity of violet flowers was one order of magnitude higher than that of white flowers (Figure 2). 

OPLS-DA was then used in the statistical analysis of the 37 compounds. As shown in Figure 3, the difference in anthocyanin content was shown to significantly affect the formation of white flowers and violet flowers. Petunidin and malvidin were observed to be the two most significantly differing anthocyanins, which were delphinidin methylated derivatives (Figure 3). By conducting an analysis on the influence of coefficient plots on violet flowers, two derivatives of delphinidin were noted to rank third and fourth, followed by three derivatives of cyanidin, which ranked fifth to seventh (Figure 3).

By comparing the differences in compounds between white flowers and violet flowers, 12 of the 13 compounds with a relatively high content of white flowers were noted to be flavonoids. Moreover, according to the mass spectrum of Figure 2, cynaroside and kaempferol in white flowers were shown to have prominent mass spectrum detection peaks, while the coefficient plot analysis demonstrated that their effects on white flowers were ranked in the top two. In light of the above analysis, the metabolic mechanism of white flower formation was preliminarily found to be due to the decrease in anthocyanin synthesis and accumulation in the delphinidin and cyanidin pathways, resulting in the anthocyanin content of white flowers to be much smaller than that of violet flowers. The white flower contains the main anthocyanin compounds, such as delphinidin derivatives and cyanidin derivatives in the violet flower, and also contains pelargonidin and its derivatives. If white flowers are caused by mutations in one or more synthesis genes, derivatives of compounds such as cyanidin, delphinidin, or pelargonidin would not be detected due to disruption of the anthocyanin synthesis. Accordingly, the flower pigment pathway of the white flower was considered to be complete, where it was ruled out that the color of the white flower was due to a mutation of the flower pigment synthesis genes. Hence, the formation of white flowers may be due to the differential expression of transcriptional regulation of flower pigment synthesis genes.

### 2.3. Full-Length Transcriptomics Analysis of White and Violet Flowers of M. latifolium

PacBio full-length transcriptome sequencing of white and violet tepals produced 40 Gb of clean data. Here, a total of 61,009 non-redundant transcripts were obtained by removing redundancies from all sequences. In the functional annotation of these non-redundant transcripts, a total of 57,153 were annotated, of which 56,937 were annotated by the nr database, 55,580 by eggNOG, 42,438 by Swiss-Prot, 46,151 by Pfam, 37,679 by KOG, 27,177 by KEGG, 35,380 by GO, and 24,506 by COG (Figure 4A). Complete ORF prediction was performed on all non-redundant transcripts, after which 42,819 were obtained. The length distribution of the encoded protein sequence is shown in Figure 4C. A total of 5454 transcription factors were obtained from all annotated transcription factors (Figure 4D). According to the statistics of nr homologous species, *M. latifolium* had the highest homology with *Asparagus officinalis*, which had 37,112 transcript annotations and accounted for 65.22% of all non-redundant transcripts (Figure 4B, Appendix A).

### 2.4. Differential Transcriptomics Analysis of White and Violet Flowers of M. latifolium

Second-generation transcriptomics carried out three biologically repeated sequencings of white and violet flowers, with each obtaining 18 Gb of clean data. The 61,009 non-redundant transcripts obtained by PacBio full-length transcriptomics were used as reference sequences for the expression analysis (Appendix A). Comparing all gene expression data of violet and white flowers, 12,237 differentially expressed genes were obtained, including 5540 upregulated genes and 6697 downregulated genes (Figure 5A). All 12,237 differential transcripts were found to be annotated, and 11,496 annotation functions were obtained, including 5386 COG annotations, 7443 GO annotations, 5339 KEGG annotations, 7004 KOG annotations, 9771 Pfam annotations, 8985 Swiss-Prot annotations, 8985 Swiss-Prot annotations, 11,139 eggNOG annotations, and 11,435 NR annotations (Figure 5B). The GO annotation of 7443 transcripts was classified according to their biological processes, molecular functions and cell component, as shown in Figure 5C. KEGG pathway enrichment was then performed on the 5339 annotated transcripts. Here, the flavonoid and phenylalanine pathways were observed to be significantly enriched, ranking first and second (Figure 5D).

### 2.5. The Expression of the Flower Pigment Synthesis Pathway between White and Violet Flowers

The expression of the flower color pathway genes obtained via Pacbio full-length transcriptomics was analyzed by second-generation transcriptomics, after which the expression levels of similar genes were then combined. Here, the expression levels of the flower pigment synthesis genes in white flowers, such as *PAL*, *4CL*, *C4H*, *CHS*, *CHI*, *F3H*, *F3’H*, *F3’5’H*, *DFR*, *ANS*, and *GT*, were found to be different from violet flowers. Moreover, the total expression levels of downstream flower pigment synthesis genes *F3H*, *F3’H*, *F3′5′H*, *DFR, ANS*, and *GT* were noted to be significantly lower than those of violet flowers (Figure 6, Appendix A). It seems that the causative gene, if any, is somewhere at the very beginning of the metabolic pathway, as in most of steps the expression is lower than in violet ones. However, in fact, pigment accumulation depends on the high expression of the whole synthesis genes. It is almost impossible to reduce anthocyanin synthesis due to mutations in all pigment genes, since white and violet flowers belong to the same raceme and have the same genome. Accordingly, the differential expression of these pigment synthesis genes is more likely to be caused by differential regulation of transcription factors.

### 2.6. Differential Expression Analysis and Subcellular Localization of MaMYB113a/b

The differential expression of flower pigment synthesis genes was speculated to be due to the regulatory differences of transcription factors. Therefore, the expression of all *MYB*s annotated transcripts was analyzed. Ten different *MYB* transcripts were subsequently obtained by screening differential transcripts with log2Fold change >2 (Figure 7B). One of the most significant differences was annotated as *AtMYB113*, which depicted homology to *Arabidopsis thaliana* pigment-positive regulatory genes. Cloning and sequencing the gene showed the presence of a 15 base insertion in the gene, which did not cause subsequent frameshift mutations. After comparing all MYBs of *M. latifolium* with all MYBs of *A. thaliana*, they were found to be most similar to *AtMYB113* and were then named *MaMYB113*. Meanwhile, a shorter sequence with a CDS length of 669 bp was named *MaMYB113a* (NCBI accession number: OP852660), while the longer sequence with a CDS length of 684 bp was named *MaMYB113b* (NCBI accession number: OP852661). Subcellular localization of these transcription factors demonstrated their localization in the nucleus. Real-time quantitative PCR analysis of *MaMYB113a/b* expression of the white and violet flowers showed that expression was significantly lower in white flowers.

### 2.7. Phylogenetic Tree Analysis and Multiple Sequence Comparison of MaMYB113a/b

*MaMYB113a/b*, as well as *AtMYB113*, *AtMYB75* (*PAP1*), *AtMYB90* (*PAP2*), and *AtMYB114*, were found to be clustered the closest by constructing a phylogenetic tree cluster of *MaMYB113a/b* and all *A. thaliana MYBs* (Figure 8A). Four *A. thaliana* flower color-related MYB transcription factors, *MaMYB113a/b*, and other species of grape hyacinth *MaMYBAN2* and *MaMybA* were reported, with a multiple sequence alignment of amino acid sequences demonstrating that *MaMYB113a/b* was homologous to pigment-positive regulatory *MYBs* transcription factors in conserved structures (Figure 8B). The 15 bp insertion sequence of *MaMYB113* gene did not cause an amino acid frameshift, and no deletions of key conserved sites were noted.

### 2.8. Transformation of MaMYB113a and MaMYB113b in Tobacco

Overexpression vectors pCAMBLA1304: *MaMYB113a* and pCAMBLA1304: *MaMYB113b*, as well as the empty vector (pCAMBLA1304), were transformed into the *Nicotiana tabacum*’ NC89’ leaf using an Agrobacterium–mediated method. Transformation empty vector (CK) tobacco and wild tobacco (WT) were used as the controls. More than ten transgenic tobacco plants were thus obtained. The leaves of transgenic tobacco plants containing *MaMYB113a* or *MaMYB113b* were found to have obvious red spots (Figure 9D). Moreover, transgenic tobacco plants with *MaMYB113a* gene were named as MaMYB113A–1, MaMYB113A–2, and MaMYB113A–3, respectively. Meanwhile, transgenic tobacco plants with *MaMYB113b* gene were named as MaMYB113B–1, MaMYB113B–2, and MaMYB113B–3, respectively. The PCR assay showed clear target bands compared with that of CK and WT (Figure 9B). The quantitative expression analysis of flower pigment synthesis genes in *MaMYB113a/b* transgenic tobacco illustrated that *NtF3H*, *NtF3′H*, *NtF3′5′H*, *NtDFR*, *NtANS*, and *NtUFGT* were significantly upregulated (Figure 9C).

A total of 45 compounds were detected in *MaMYB113a/b* transformed, WT and CK tobacco by targeted mass spectrometry of 72 flower color related standards (Figure 10, Appendix A). The content of 19 compounds in *MaMYB113a/b*-transformed tobacco was found to increase significantly, of which 11 anthocyanin compounds were shown to increase significantly. Mass spectra showed that cyanidin–3–*O*–rutinoside and delphinidin–3–*O*–rutinoside were the most significant anthocyanins. The expression of genes related to flower pigment synthesis and types of flower color-related compounds was noted to be basically the same in tobacco transformed with *MaMYB113a* and *MaMYB113b*, respectively. The results preliminary showed that *MaMYB113a* and *MaMYB113b* had the same effect on the positive regulation of tobacco pigment color. Overall, the overexpression of *MaMYB113a/b* was observed to promote anthocyanin synthesis in tobacco.

## 3. Discussion

*Muscari latifolium* has garnered increased interest in research due to its blue–purple double color variation. Mori speculated that the upper blue flower was mainly delphinidin, while the lower blue–violet flower was a mixture of cyanidin and delphinidin [15]. Qi maintained this conclusion using HPLC, where it was shown that the light blue upper flower of *M. latifolium* contains lower delphinidin, without cyanidin, petunidin, or malvidin; however, petunidin and malvidin were not detected in the lower flowers [16]. Subsequently, Wang and Lou used untargeted mass spectrometry to detect cyanidin in the upper flowers, though no delphinidin derivatives were detected. Meanwhile, petunidin derivatives and malvidin derivatives were detected in the lower flowers [17,18].

This study collected and examined 72 flower color-related standards and employed a one-by-one targeted detection in order to accurately characterize the newly discovered white–violet mutant of *M. latifolium*. A total of 37 compounds were thus identified in the two colors of the flowers, which all contained derivatives of delphinidin, petunidin, pelargonidin, cyanidin, and peonidin. The corresponding findings showed that white flowers and violet flowers contained 25 common compounds, different from previous conclusions. This finding may further enrich the species of flower color substances of *M. latifolium.*

After excluding the possibility of pH and metal elements causing the white–violet color, full-length transcriptomics was used to analyze the expression of flower pigment synthesis genes. Since full-length transcriptome data were not reported for grape hyacinth, PacBio sequencing was utilized in order to obtain 61,009 non-redundant transcripts. The comparative expression analysis of *PAL*, *4CL*, *C4H*, *CHS*, *CHI*, *F3H*, *F3’H*, *F3’5’H*, *DFR*, *ANS*, and *GT* in the flower pigment synthesis pathway showed that these genes were all present in white flowers and violet flowers; however, the overall expression of *F3H*, *F3’H*, *F3’5’H*, *DFR*, *ANS*, and *GT* in white flowers was found to be significantly lower than that in violet flowers. Thus, the formation of white flowers was speculated to not be due to the deletion of genes, which also was not consistent with Mori’s conclusion in that the difference between blue and white flowers was due to the differential expression of a single gene, *F3’5’H* [21].

In the past, two R2R3–MYB transcription factors, *MaMybA* and *MaAN2*, have been obtained and validated in *M. armeniacum*, which are known to function in promoting flower pigment accumulation in tobacco [19,20]. Here, all *MYBs* of the full-length non-redundant transcripts of *M. latifolium* were analyzed, where no *MaMybA* and *MaAN2* genes were detected in *M. latifolium*. However, among all *MYBs* transcription factors in white–violet flowers, transcription factor gene *MaMYB113* with a significant difference in expression was observed, which was highly expressed in violet flowers and lowly expressed in white flowers. Functional annotation was noted to be highly homologous to *AtMYB113* [22]. Moreover, the phylogenetic tree analysis showed that MaMYB113a/b had the highest homology with four well-known transcription factors AtMYB113, AtMYB75 (PAP1), AtMYB90 (PAP2), and AtMYB114, which significantly promoted flower pigment synthesis in *A. thaliana* [23,24]. Subsequent transgenic tobacco confirmed this hypothesis in that overexpression of *MaMYB113a/b* can play a role in promoting the flower pigment synthesis pathway (Figure 9).

Meanwhile, we also found that the flowering development process of the white–violet bicolor variation *M.latifolium* was not affected even if there was differential expression of M*YB*s. However, the growth and development of *35S: MaMYB113*-transformed tobacco was slower than that of wild type tobacco. This is because the accumulation of large amounts of anthocyanin in leaves led to photosynthesis being repressed. Therefore, for transgenic engineering plants, effective expression of *MYB*s at the appropriate time and tissue sites, rather than unlimited arbitrary expression, may have better practical value. In addition to preliminarily verifying that *MaMYB113a/b* can promote the synthesis of flower color genes, three topics must be studied further. First, other than *MaMYB113a/b*, certain significantly differentially expressed *MYBs* were screened (Figure 7B). Do these *MYBs* all play an important role in the double-color variation of white and violet? Second, flower color regulation may be dependent on MYB alone while forming MYB–bHLH, MYB–WD40, or MYB–bHLH–WD40 protein complexes in order to regulate flower pigment synthesis genes [7,8]. Hence, does MaMYB113a/b also regulate flower pigment synthesis by interacting with other bHLHs and WD40s? Finally, how are *MaMYB113a/b* differentially expressed in white and violet flowers in the *raceme*? Compared with traditional breeding methods such as hybrid breeding and mutation breeding, molecular breeding via genetic engineering technology is faster and more accurate, which can accelerate the breeding process and cultivate new varieties with excellent bicolor flower traits in the future. [1]. Therefore, the establishment of an efficient molecular mechanism transgenic system of grape hyacinth will also be very important.

In light of the aforementioned questions, we aim to continue to explore the mechanism of *M. latifolium* white–violet double color mutation to provide a theoretical basis for the cultivation of double color flowers.

## 4. Materials and Methods

### 4.1. Plant Materials

*Muscari latifolium* cultivar ‘Latifolium’ were preserved in the nursery of the College of Forestry, Northwest A&F University. In terms of the new white and violet bicolor mutant, all white and violet flowers from early non-opening to full bloom were collected.

### 4.2. Microscopic Observation

This study was conducted during the time period in the two weeks of flowering from late March and early April. The following method was used: (1) the whole *M. latifolium* was dug up with the planting basin to the Leica microscope room; (2) the floating soil impurities on the tepal surface were blown off using a low-speed blower; (3) a white floret and violet floret were picked and were cut in half-cut with a super Gillette blade; (4) the tepals were quickly cut with two blades on the water-dropping glass slide, and the thinnest tepal cross-section was picked. The cover glass was immediately observed under a Leica microscope(Wetzlar, Hessen, Germany).

### 4.3. Puncture pH of Tepals

The time was selected at 8 a.m. every day during the two-week flowering period at the end of March and early April. The main methods performed were: (1) Thermo Scientific™ Orion™ 9863BN Micro pH Electrode (Waltham, MA, USA) was calibrated from acid, neutral, and alkaline in turn by using the pH calibration solution; (2) ultrapure water blank control was measured; (3) tepal cells were punctured assisted by Leica™ stereomicroscope(Wetzlar, Hessen, Germany) and their value was recorded; (4) ultrapure water was recalibrated and measured; (5) steps 1–4 were then repeated, and new tepal cells were re-punctured to record the pH. Then, 9 different puncture cell pH values were recorded for 3 different single flowers of white flowers and violet flowers, respectively.

### 4.4. Mass Spectrometry Analysis of Metal Elements in Tepals

The following procedure was performed: (1) tepal or leaf samples were dehydrated by low temperature drying; (2) 0.2 g dry weight was weighed to a cleaned polytetrafluoroethylene (PTFE) tube, adding 6 mL nitric acid and 1 mL hydrogen peroxide for microwave digestion, with the PTFE tube soaked in nitric acid three times prior to the operation; (3) the digestion solution was placed on the heating deacidification machine and was manually operated, after which the volatile digestion solution was heated to below 0.5 mL; (4) the PTFE tubes were then cleaned with ultrapure water, all in 25 mL bottles.

The following Thermo Scientific ^TM^ iCAP RQ ICP-MS(Waltham, MA, USA) preparation steps were performed: (1) start-up, ignition and pressure water circulation was inspected; (2) 12 kinds of standard concentration gradient were collected, after which the standard curve was drawn; (3) the pipeline was cleaned; (4) for each needle sample, the pipeline was cleaned once until all samples were measured. The real-time detection concentration feedback unit was in ppm. The corresponding instrument operating parameters were adhered to: forward power: 1500 W; atomizing gas flow: 0.9 L/min; auxiliary gas flow: 0.8 L/min; cooling gas flow: 14 L/min; helium: 5 mL/min; sample uptake time: 45 s; sample wash time: 45 s; number of pints per week: 1; number of repeats per sample: 3; and acquisition time: 3 min [25].

### 4.5. Targeted Mass Spectrometry Analysis of Flower Color Related Compounds

The following sample preparation steps were carried out: (1) the day before the mass spectrometry test, a fresh weight of 200 mg was weighed and dissolved in 1 mL of methanol acetonitrile ratio with a pH value of neutral at 0 °C pre-cooled extract; (2) zirconium beads were added in a 2 mL test tube containing the sample extract; (3) in the 0 °C refrigerator, it was broken with a tissue grinder, with a breaking time of 3 min; (4) in the 0 °C freezer, machine extraction was performed for 18 h; (5) using a 0.1 micron organic phase filter, the head filter was extracted twice; (6) the filtrate was transferred to 1.5 mL brown mass spectrometry sample bottle, stored in ice boxes, and left for testing.

The following parameters were used: UPLC preparation: phase A 0.1% formic acid water (pumping); and phase B 0.1% formic acid acetonitrile (pumping). In regard to the flow phase parameters, the initial A:B was 85:15; linear gradient elution for 3 min until A:B was 0:100; B phase was maintained at 100% for 1 min; A:B was restored to the initial ratio 85:15; and column equilibrium was attained for 1 min. The flow rate was 0.5 mL/min; the initial pressure of the mobile phase was 318.5 bar; and the loading volume of the automatic sampler was 1 μL. The column type was a C18 2.7 µm × 100 mm UPLC column.

Mass spectrometry analysis: Agilent 1290–6460UPLC–QQQMS/MS(Santa Clara, CA, USA) was used to establish a targeted flower color related standard method according to the following steps: (1) Qualitative analysis mass spectrometry software operation(Santa Clara, CA, USA) was used, with an initial tuning mass spectrometry triple quadrupole detection device set to tuning type QQQ, autotune automatic, and positive ionization scanning mode; (2) a standard sample concentration range of 100–1000 pg/μL was prepared, noting that an excessive concentration will lead to machine error and sensor response problems; (3) the MS2 scan single sample mode and tuning file selection Atune TUNE XML were used, where a single parent ion full scan was performed, and the main peak parent ion scanned was analyzed and recorded; (4) a batch working file worklist was established, after which the main peak parent ion determined in the previous step was selected, the Fragmenator voltage value of each line of the batch working file was changed, and the optimal parent ion transmission voltage was checked in an arithmetic arrangement, ensuring that the sample was transferred to the mass spectrum with the maximum intensity of the parent ion; (5) a new batch working file worklist was established, the fragmentation voltage value of each line of the batch working file was changed, and the optimal collision energy of the parent ion was tested, in which the energy was entered in an arithmetic progression until all parent ions in the detection map were broken and formed characteristic sub-ion fragments, while the most responsive sub-ion fragment values and auxiliary identification fragment values were detected and analyzed; (6) MRM was optimized, a new batch worksheet worklist was created, and the parameters were adjusted to MRM optimal monitoring; (7) tandem Agilent 1290–UPLC was used, which recorded in liquid phase conditions of the main peak parent ion and MRM delay state fragment ion detection time. All standards repeated the above steps until 72 standards were measured to form a mass spectrometry method file. See Appendix A for further information.

Multivariate statistical data analysis and OPLS–DA analysis was performed with SIMCA-P version 14.1 (Umetrics, Umeå, Sweden).

### 4.6. RNA Extraction and cDNA Synthesis

White and violet flowers of *M. latifolium* were rapidly frozen in liquid nitrogen, collected at successive stages during 2 weeks of heading and flowering, respectively, and used to assess RNA integrity according to 28S requirements: 18S ≥ 1.4, RNA integrity (RIN) ≥ 9. Total RNA samples from three white flowers and three violet flowers were then combined in equal amounts for cDNA synthesis and SMRTbell library construction. Clontech SMARTer PCR cDNA synthesis kit (Takara Biotechnology, Dalian, Liaoning, China) was used for cDNA synthesis, and PacBio SMRTbell template preparation kit (PacBio, Menlo Park, CA, USA) was used to construct the SMRTbell library. Using the NEB next Ultra RNA Library Prep Kit (NEB, Ipswich, MA, USA), a total of 2 μg RNA was used to produce independent sequencing libraries for 6 samples. The library was sequenced using the Illumina HiSeq X (San Diego, CA, USA). system in a 150 bp paired-end mode.

### 4.7. Transcriptome Library Construction, Sequencing and Data Analysis

Quality control of raw data was performed in order to avoid conducting an inaccurate analysis of the subsequent information. Following data quality control, the amount of data of the sub-reading was then calculated. The common sequence generated by multiple sub-reading sequences was found to be CCS reading, which was further corrected to obtain HQ isomers (accuracy greater than 0.99) and LQ isomers. Homologous sequences with a large number of redundant sequences were clustered together using an isotype-level clustering algorithm called ‘Iterative Clustering for Error Correction’ (ICE) to obtain a new common isotype [26]. Using the common isotype obtained after ICE as a reference sequence, the short reading sequences obtained from the six Illumina RNA-seq libraries were then compared with the redundantly removed reference sequences of the full-length sequences obtained using PacBio. The expression level of the transcript was quantified using RSEM [27] by applying the location information of mapped reads on the third-generation transcripts. The number of fragments extracted from a transcript was then compared to the amount of sequencing data (or mapped data), length of transcripts, and expression level of transcripts. In order to make the number of fragments truly reflect the expression level of transcripts, the number of mapped reads and length of transcripts in the sample were normalized. FPKM was used as an indicator to measure transcript or gene expression levels, and a differential gene expression analysis between different treatment groups was performed using DEGSeq [28]. Those having a gene *p*-value of 0.05 and satisfying the condition of log2 fold change rate ≥1 were identified to be DEGs. The obtained non-redundant transcript sequences were then aligned with NR [29], Swiss-Prot [30], GO [31], COG [32], KOG [33], Pfam [34], and KEGG [35] databases using BLAST [36] (version 2.2.26) in order to obtain the annotation information for the transcripts.

### 4.8. Phylogenetic Tree Analysis and Protein Sequence Alignment

All MYBs of *Arabidopsis* were obtained from the TAIR database (http://www.arabidopsis.org/, accessed on 1 December 2022). Multiple sequence alignment was performed using Clustal Omega [37]. MEGA7 builds a phylogenetic tree, and the maximum likelihood was constructed with 1000 replications of bootstrap. The obtained phylogenetic tree was optimized using iTOL [38].

### 4.9. MaMYB113a/b Subcellular Localisation

The *MaMYB113a/b* CDS gene sequence was then removed from the termination codon, and the homologous recombination arm was added to the design primers for PCR. Following electrophoresis, the gel was purified, and the pCAMBIA2300-GFP vector was digested with *BamH*I/*Sal*I. Homologous recombination was then performed and transformed into *Escherichia coli* DH5α. After coating the plate and selecting monoclonal sequencing, the subcellular localization vector of the correct sequence was obtained.

pCAMBIA2300-MYB113a-GFP and pCAMBIA2300-MYB113b-GFP were used for heat shock transformation of GV3101 *Agrobacterium* competent cells, respectively. The plate was coated to pick up monoclonal after LB activation culture, which used a 50 mL centrifuge tube and 5 mL LB overnight shaker culture and centrifugal collection of bacterial liquid, after which centrifugal resuspension was performed twice to collect the washing bacterial liquid. Finally, shock resuspension in MES buffer (200 μM acetosyringone, 0.01 M magnesium chloride hexahydrate, 0.01 M MES) was performed, and the bacterial liquid OD_600_ concentration was adjusted to about 0.8. It was left to stand in the dark for two hours, after which the back of the tobacco leaves was injected with a clean 1 mL syringe. Following injection, it was kept in dark incubation for two days. The fluorescence intensity was then observed via Leica™ laser confocal microscopy(Wetzlar, Hessen, Germany) with inverted leaf sections.

### 4.10. Tobacco Transformation

The vector of the pCAMBIA1304 hygromycin-resistant tobacco transformation vector was preserved in our laboratory. The pCAMBIA1304-MaMYB113a and pCAMBIA1304-MaMYB113b *Agrobacterium* infection transformation vectors were obtained by *Nco*I and *Bgl*I double digestion and homologous arm recombination *MaMYB113a/b*. The method transformation was adopted from previous studies that were carried out in this laboratory [18,19].

### 4.11. Real Time quantitative PCR (RT-qPCR)

RNA Kit (Omega, Knox, GA, USA) was used to extract total RNA, and 1μg RNA was used to synthesize cDNA using the PrimeScript™ RT reagent Kit with gDNA Eraser (Takara, Dalian, China). qRT-PCR was performed on an iQ5 RT-PCR instrument (Bio-Rad, Hercules, CA, USA) using the qPCR SYBR ^®^ Green Master Mix (Vazyme, Nanjing, Jiangsu, China). The *MaActin* gene was selected as the internal reference gene, and *NtTubA1* gene was selected as the internal reference gene of tobacco [39]. The primers for RT-qPCR are shown in Appendix A. The relative expression of gene was calculated using the 2^−ΔΔCt^ method [40]. All analyses were conducted by carrying out three independent experiments.

## Figures and Tables

**Figure 1 ijms-24-05044-f001:**
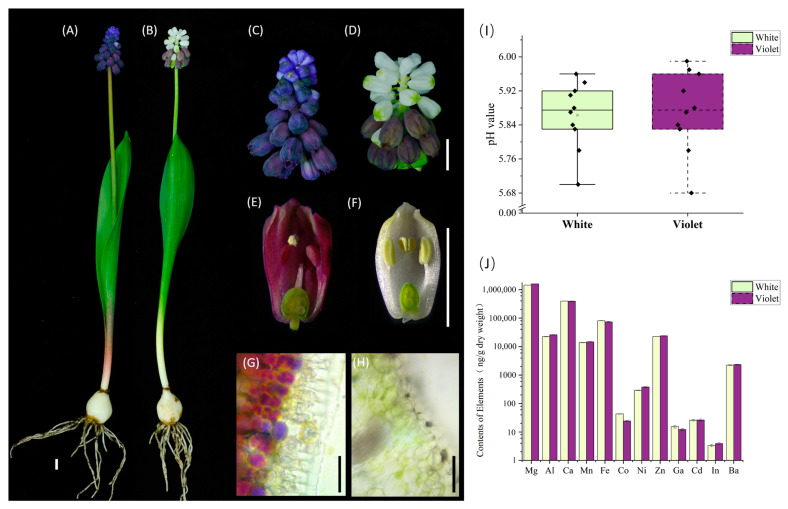
Phenotypic and ionomics analysis of *M. latifolium* white–violet bicolor mutant. (**A**) Natural *M. latifolium* whole plant with blue–violet inflorescence. (**B**) Mutant *M. latifolium* white–violet bicolor inflorescence of whole plant. (**C**) Natural *M. latifolium* blue–violet inflorescence phenotype. (**D**) Mutant *M. latifolium* white–violet inflorescence phenotype. (**E**) Natural *M. latifolium* blue–violet single flower anatomy. (**F**) Mutant *M. latifolium* white single flower anatomy. (**G**) Microscopic anatomy of natural *M. latifolium* blue–violet flowers. (**H**) Microscopic anatomy of the mutant *M. latifolium* white flower. (**I**) White and violet tepals pH value. (**J**) The contents of 12 blue–violet related metal elements in white and violet flowers. Bars: (**A**–**F**) 5 mm; (**G**–**H**) 50 μm.

**Figure 2 ijms-24-05044-f002:**
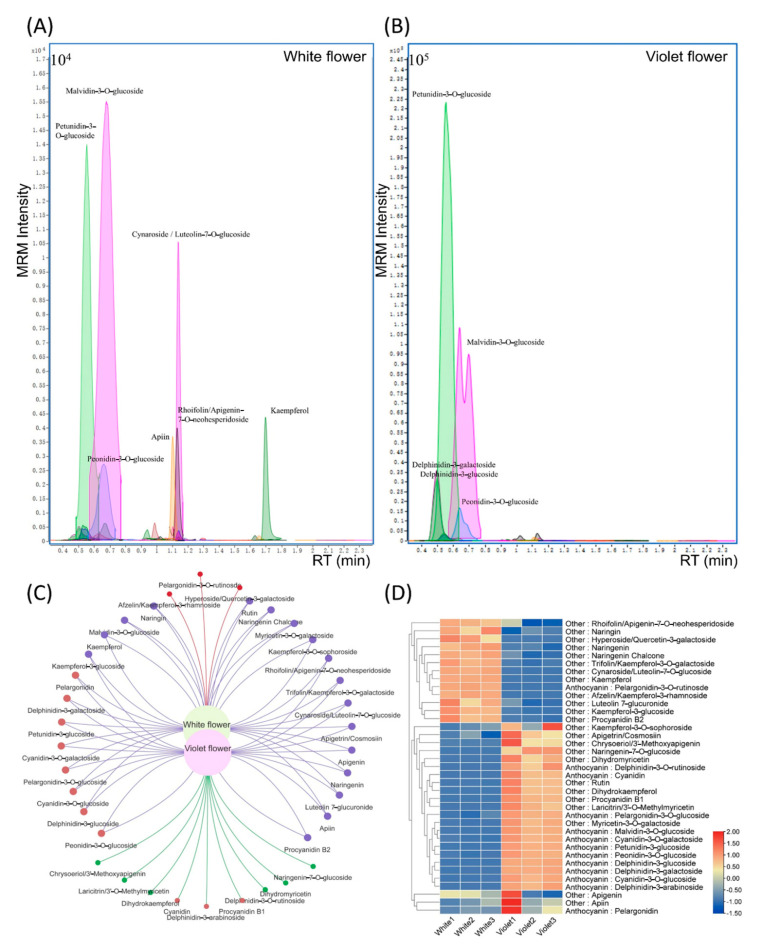
Targeted metabolomics analysis of white and violet flowers of *M. latifolium*. (**A**) MRM detection of targeted color−related compounds in white flowers. (**B**) MRM detection of targeted color−related compounds in violet flowers. (**C**) Venn diagram of compound types of white flowers and violet flowers. (**D**) Heat map of compounds between white flowers and violet flowers. White1: white flower sample 1; White2: white flower sample 2; White3: white flower sample 3; Violet1: violet flower sample 1; Violet2: violet flower sample 2; Violet3: violet flower sample 3.

**Figure 3 ijms-24-05044-f003:**
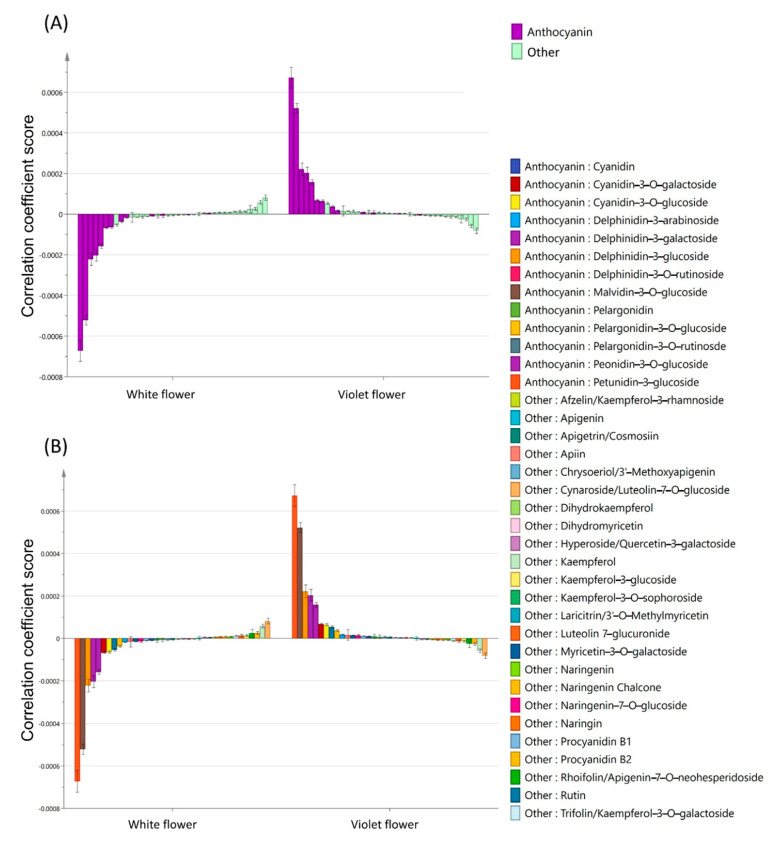
OPLS−DA analysis of compounds differences between white flowers and violet flowers. (**A**) Coefficient plots of the OPLS−DA model of the compounds type analysis between white and violet flowers. (**B**) Coefficient plots of the OPLS−DA model of the compounds analysis between white and violet flowers.

**Figure 4 ijms-24-05044-f004:**
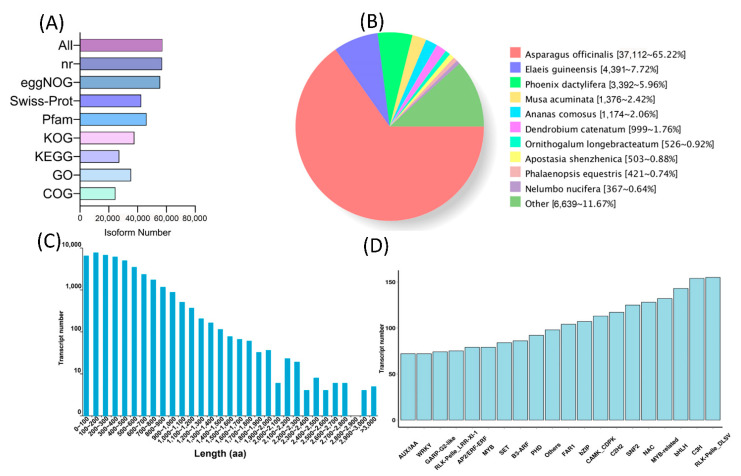
Full-length transcriptomics analysis of white and violet flowers of *M. latifolium*. (**A**) Multi-database annotation of all non-redundant transcripts. (**B**) Nr annotation species classification statistics of all non-redundant transcripts. (**C**) The predicted CDS–encoded protein length distribution of all non-redundant transcripts. (**D**) The type and number of transcription factors for all non-redundant transcripts.

**Figure 5 ijms-24-05044-f005:**
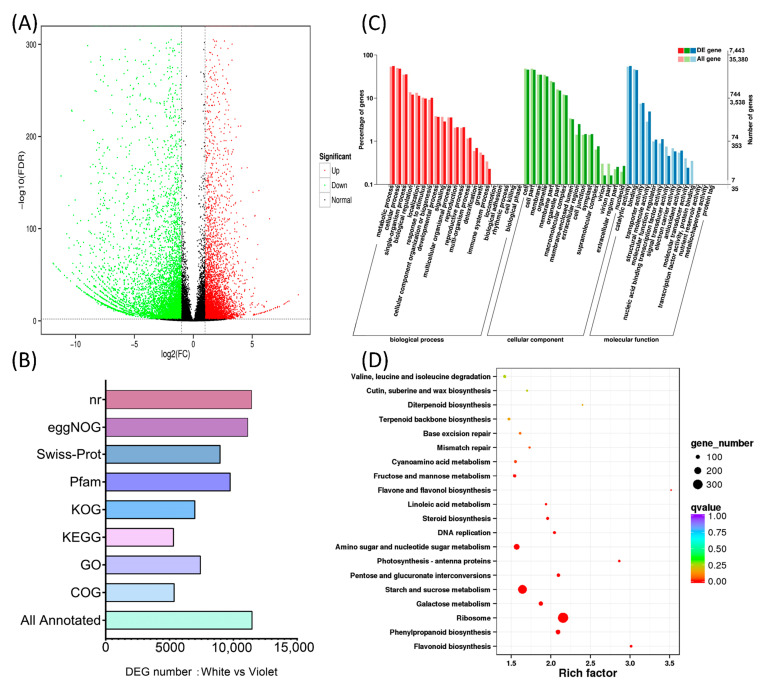
Differential transcriptomics analysis of white and violet flowers. (**A**) Volcano diagram of the transcript differential expression between white and violet flowers. (**B**) Multi-database annotation of differentially expressed transcripts between white and violet flowers. (**C**) GO annotation of differentially expressed transcripts between white flowers and violet flowers. (**D**) KEGG pathway enrichment annotation of differentially expressed transcripts between white and violet flowers.

**Figure 6 ijms-24-05044-f006:**
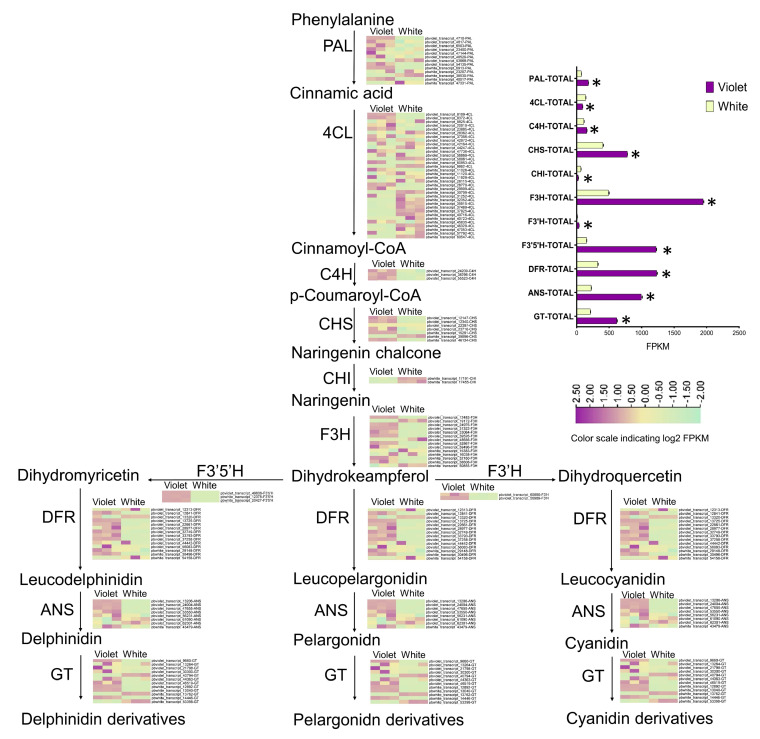
The expression map of flower pigment synthesis pathway between white and violet flowers. Student’s *T* test is denoted by *, *p* < 0.05.

**Figure 7 ijms-24-05044-f007:**
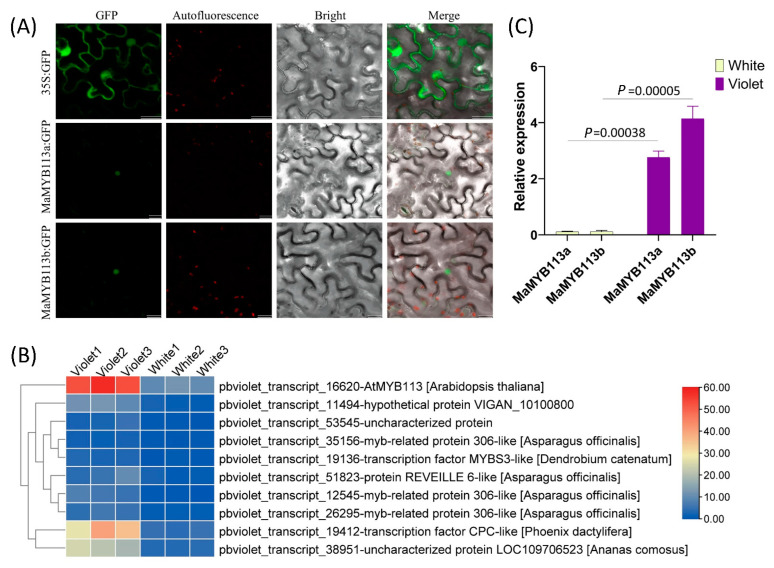
Differential expression analysis and subcellular localization of *MaMYB113a/b*. (**A**) Subcellular localization of *MaMYB113a/b* in tobacco cells. Bars: 20 μm (**B**) Expression heat map of *MaMYB113* in white and violet flowers in all *MYB* differentially expressed transcripts selected with log2Fold change >2. (**C**) Relative expression analysis of *MaMYB113a* and *MaMYB113b* in white and violet flowers by 2^−ΔΔCt^.

**Figure 8 ijms-24-05044-f008:**
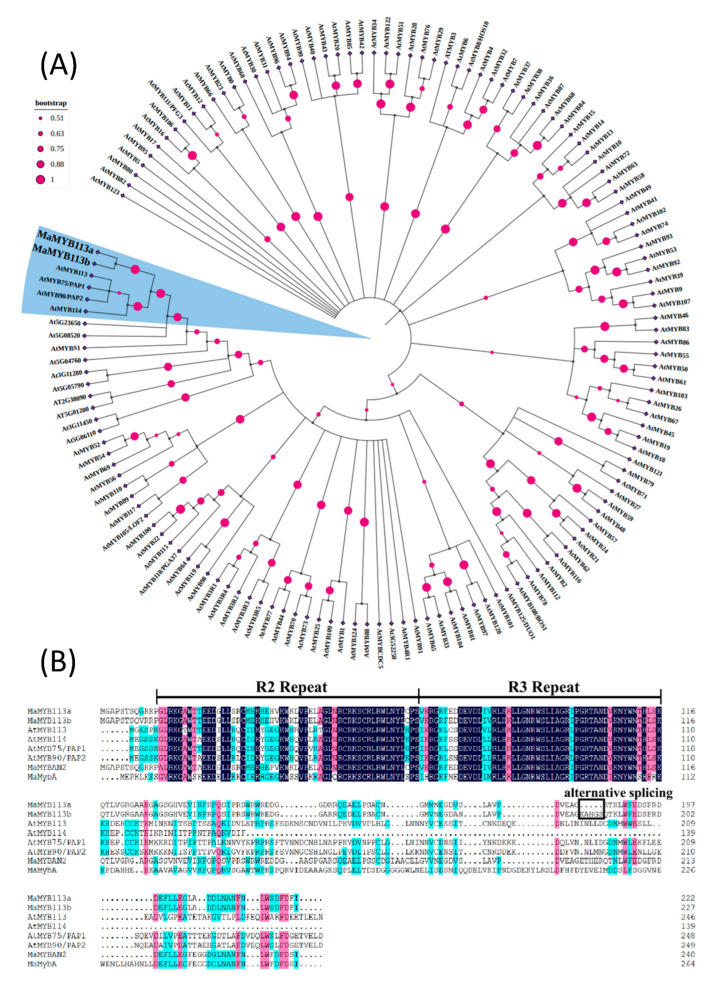
Phylogenetic tree analysis and multiple sequence comparison of *MaMYB113a* and *MaMYB113b*. (**A**) Phylogenetic analysis of *MaMYB113a* and *MaMYB113b* together with all *MYB*s of *A. thaliana*. (**B**) Multiple sequence alignment of *MaMYB113a* and *MaMYB113b* with similar *MYBs* from Arabidopsis and grape hyacinth. Maximum likelihood trees were constructed with 1000 replications of bootstrap.

**Figure 9 ijms-24-05044-f009:**
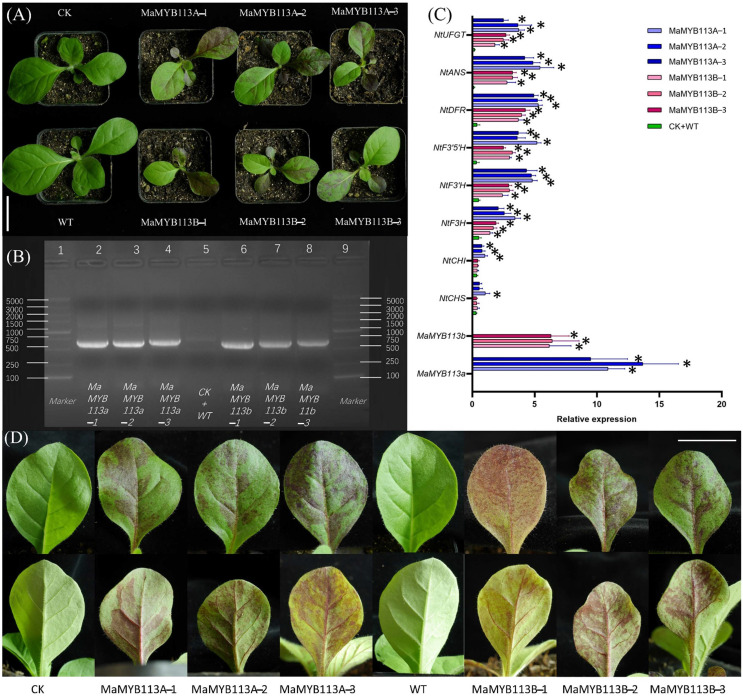
Stable transformation of *MaMYB113a* and *MaMYB113b* in tobacco. (**A**) Phenotypic analysis of *MaMYB113a* and *MaMYB113b* after transformation into tobacco; CK: empty vector; WT: wild type. (**B**) PCR analysis of *MYB113a/b* was performed on tobacco, respectively. Electrophoresis lane 1 and 9: marker; lane 2: MaMYB113A–1; lane 3: MYB113A–2; lane 4: MYB113A–3; lane 5: CK + WT; lane 6: MYB113B–1; lane 7: MYB113B–2; lane 8: MYB113B–3. (**C**) Relative expression analysis of overexpression of *MaMYB113a/b* on expression of flower color genes in tobacco leaves by 2^−ΔΔCt^. Student’s *T* test is denoted by *, *p* < 0.05. (**D**) Front and back of leaf phenotype of transgenic tobacco and control tobacco. The first row is the leaf front phenotype, and the second row is the leaf back phenotype. Bars: 2 cm.

**Figure 10 ijms-24-05044-f010:**
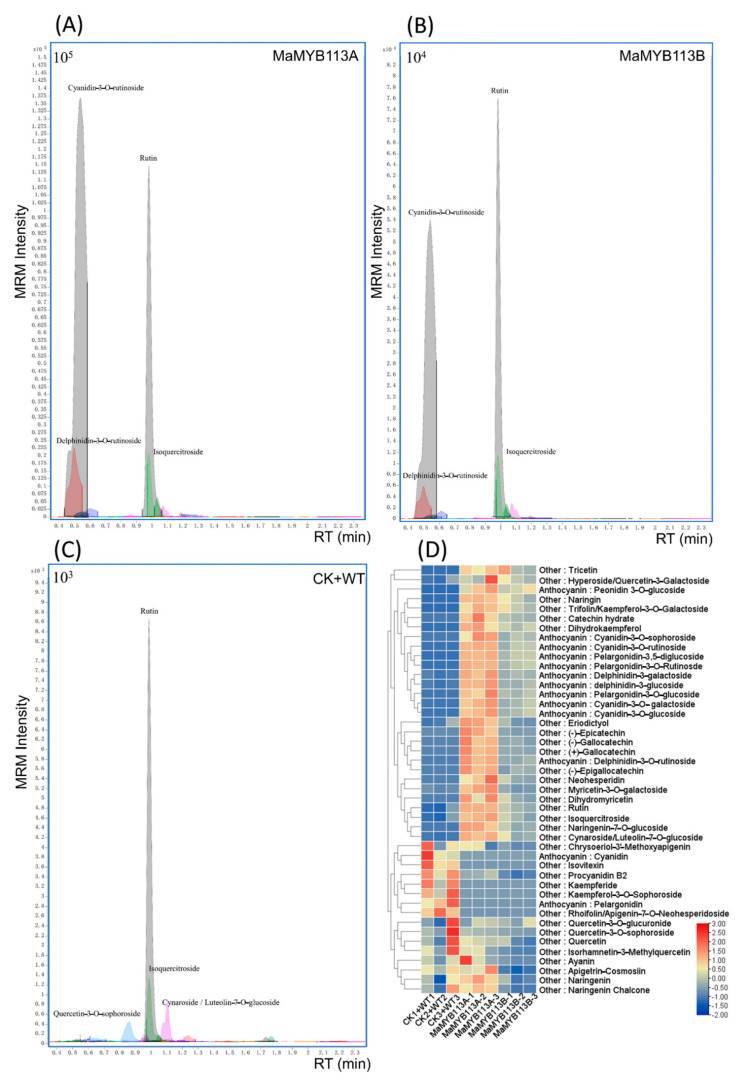
Targeted metabolomics analysis of tobacco leaves overexpressing *MaMYB113a* and *MaMYB113b*. (**A**) MRM detection of targeted color-related compounds in tobacco leaves overexpressing *MaMYB113a*. (**B**) MRM detection of targeted color-related compounds tobacco leaves overexpressing *MaMYB113b*. (**C**) MRM detection of targeted color-related compounds in WT and CK tobacco leaves. (**D**) Heat map of compounds between overexpressed *MaMYB113a/b* and WT and CK tobacco leaves.

## Data Availability

The data of this study have been deposited into CNGB Sequence Archive (CNSA) of China National GeneBank DataBase (CNGBdb) [41] with accession number CNP0002731.

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
