# Peer review of "Integrating Multi-Omics Analysis Reveals the Regulatory Mechanisms of White–Violet Mutant Flowers in Grape Hyacinth (Muscari latifolium)"

_ijms, 2023, doi:10.3390/ijms24055044_

Round 1

Reviewer 1 Report

The reviewed manuscript reports the results of a cross-examination of color variation in a grape hyacinth. This model is very intriquing as these variations exist not between different genotypes but within a single plant.
The authors applied numerous methods and the results obtained sound reliable, although quite cautious, as more studies are needed to finally dissect the molecular mechanism underlying this unusual phenotype.
However, I should say that the whole text requires a very deep elaboration considering its style and language. There are numerous sentences which are difficult to understant or sound not enough correct (e.g. 'synthesis of flower color' etc.). I have made some corrections and suggestions (see file attached) but there are still much room for improvement. I insist this text needs to be checked by native speaker, preferably professional biologist, prior to its resubmission. Otherwise it is very difficult to read and perceive which strikingly contrasts with a huge amounts of efforts involved in this valuable work.
Among minor although undesired flaws I may mention that authors do not italicize Latin and gene names in many cases where it is appropriate.
Table 1 doubles Figure 1I, J and is better to be placed to a supplement.
Many things need to be explained in figure legends (see attached).
This paper contains very many abbreviations of methods which in many cases remain unexplained. Please provide all methods' names in full at the very first mention. Alternatively, it is possibly worth adding a list of abbreviations at the beginning of a text, but this solution requires consulting with the Editorial Office if it meets the journal's guide for authors.
After making all needed corrections, this paper can be recommended for publication in IJMS.

Author Response

Dear Reviewer  : 
This is the final version of the revised paper using the tag 
'Track Changes'.Due to the request of native English speakers to modify the language, the reply was delayed by two days.

Reviewer 2 Report

The manuscript “Integrating Multi-Omics Analysis Reveals the Regulatory Mechanisms of White-Violet Mutant Flowers in Grape Hyacinth (Muscari latifolium)” (ijms-2078702), demonstrated the potential use of Multi-Omics analysis in Grape plants-based mechanisms of white-violet flowers. This manuscript its very interesting and show importance Molecular Plant Sciences understand to phenylpropanoids pathways.

The authors have done a large amount of work, employing various references and critical analysis based on a scientific method and structure. The introduction, M&M, the results and discussion topic is good and minor points its necessary by adjusting in English synthases. Tables and figures are good qualities. However, minor points in legends of the tables a figure is necessary adjusts. Few adjusts in format, for example, spaces between [references]. Fig. or Figures. Scientific names in italics.

My conception, the manuscript is suitable for publication, after few corrections. The structure is adequate, and the information is new and of great significance for comprehension.

In addition, I would just like to ask the authors a few questions, on small points that I was confused by or that were not written in the manuscript.

Points:

#01: There is a scope for improvement in the introduction section: a) additional emphasis on the significance of the study, b) scientific contribution of the paper in integration ornamental importance to farming? Maybe a one paragraphs with potential economic by ecotypes selected in your manuscript; although it is a molecular biology work, perhaps it could cover a wider audience of basic agronomy readers as well.

#02: How this methodology can be extended for other plants: potential challenges, advantages? Do you have any competitive disadvantages (economic, competition between species, changes in phenotypic plasticity (Figure 9)? Added in your results and discussion topic, please. For example, pathways in metabolic phenylpropanoids.

-Conclusion: What are future perspectives beyond the questions in the last few paragraphs?

Best regards

Author Response

(The authors gave the same response as above.)
